# CHST4 Gene as a Potential Predictor of Clinical Outcome in Malignant Pleural Mesothelioma

**DOI:** 10.3390/ijms25042270

**Published:** 2024-02-14

**Authors:** Shoji Okado, Taketo Kato, Yuki Hanamatsu, Ryo Emoto, Yoshito Imamura, Hiroki Watanabe, Yuta Kawasumi, Yuka Kadomatsu, Harushi Ueno, Shota Nakamura, Tetsuya Mizuno, Tamotsu Takeuchi, Shigeyuki Matsui, Toyofumi Fengshi Chen-Yoshikawa

**Affiliations:** 1Department of Thoracic Surgery, Nagoya University Graduate School of Medicine, 65 Tsurumai-cho, Showa-ku, Nagoya 466-8550, Japan; s-okado@med.nagoya-u.ac.jp (S.O.); taketokato63@gmail.com (T.K.); y-imamura@med.nagoya-u.ac.jp (Y.I.); hwatanabe@med.nagoya-u.ac.jp (H.W.); ykawasumi@med.nagoya-u.ac.jp (Y.K.); ykadomatsu@med.nagoya-u.ac.jp (Y.K.); h-ueno@med.nagoya-u.ac.jp (H.U.); shota197065@med.nagoya-u.ac.jp (S.N.); te.mizuno@med.nagoya-u.ac.jp (T.M.); 2Department of Pathology and Translational Research, Gifu University Graduate School of Medicine, Yanagido 1-1, Gifu 501-1194, Japan; yuyunyushiki@gmail.com (Y.H.); takeutit08@gmail.com (T.T.); 3Department of Biostatistics, Nagoya University Graduate School of Medicine, 65 Tsurumai-cho, Showa-ku, Nagoya 466-8550, Japan; remoto@med.nagoya-u.ac.jp (R.E.); smatsui@med.nagoya-u.ac.jp (S.M.)

**Keywords:** malignant pleural mesothelioma, CHST4, prognostic factor, immunohistochemistry, FFPE

## Abstract

Malignant pleural mesothelioma (MPM) develops primarily from asbestos exposures and has a poor prognosis. In this study, The Cancer Genome Atlas was used to perform a comprehensive survival analysis, which identified the *CHST4* gene as a potential predictor of favorable overall survival for patients with MPM. An enrichment analysis of favorable prognostic genes, including *CHST4*, showed immune-related ontological terms, whereas an analysis of unfavorable prognostic genes indicated cell-cycle-related terms. *CHST4* mRNA expression in MPM was significantly correlated with Bindea immune-gene signatures. To validate the relationship between *CHST4* expression and prognosis, we performed an immunohistochemical analysis of *CHST4* protein expression in 23 surgical specimens from surgically treated patients with MPM who achieved macroscopic complete resection. The score calculated from the proportion and intensity staining was used to compare the intensity of *CHST4* gene expression, which showed that *CHST4* expression was stronger in patients with a better postoperative prognosis. The median overall postoperative survival was 107.8 months in the high-expression-score group and 38.0 months in the low-score group (*p* = 0.044, log-rank test). Survival after recurrence was also significantly improved by *CHST4* expression. These results suggest that *CHST4* is useful as a prognostic biomarker in MPM.

## 1. Introduction

Malignant pleural mesothelioma (MPM) is a cancer that occurs in the pleura that lines the walls of the chest and lungs. The disease is caused mainly by asbestos exposure and has a very poor prognosis even in early disease stages, with a median survival of only 14 months [1]. Although the most effective treatment is surgery, many patients are not candidates for surgical resection because they have an advanced disease stage, so chemotherapy is instead the preferred treatment [2,3]. Combination immunotherapy with ipilimumab and nivolumab has emerged as an effective treatment modality for MPM during the last decade. However, the factors related to immunotherapeutic responses remain elusive [4].

Given the malignant nature of MPM and the limited number of patients eligible for surgery, there is a need to elucidate the pathogenesis and identify approaches that may lead to effective treatment. Genome-wide gene expression resources, such as The Cancer Genome Atlas (TCGA), have been developed to aid research and discovery of large cohorts worldwide to better understand the effect of tumor genetic composition [5]. Although extensive research has been conducted on the mechanisms underlying the incidence and progression of MPM, beneficial therapies or diagnostic tools for use in clinical practice are still under investigation or in preclinical trials.

*CHST4* is a sulfotransferase involved in the SELL/L-selectin ligand biosynthesis pathway and is expressed in normal tissues of the liver, gallbladder, pancreas, and oviduct. This sulfotransferase has an important role in lymphocyte homing. The homing of lymphocytes to secondary lymphoid organs requires the interaction of L-selectin on lymphocytes with sulfated glycoprotein ligands on high endothelial venules (HEVs) in lymph nodes. *CHST4* localizes to HEVs in lymph nodes and, as a sulfotransferase, is involved in the generation of the MECA-79 epitope induced in chronic inflammatory tissues and malignant tumors [6,7,8]. Tissue injury caused by the spread of this epitope leads to the formation of various lesions; Longshan et al. reported that *CHST4* may recruit immune cells into the tumor microenvironment and prevent the progression of hepatitis B virus-related hepatocellular carcinoma [9]. Various solid tumors, such as mucinous adenocarcinoma, cervical cancer, and uterine cancer, show high levels of *CHST4*, suggesting its usefulness as a biomarker in malignant tumors [10,11].

The present study’s aim was to use the TCGA MPM dataset to perform a comprehensive survival analysis. The analysis identified *CHST4* among all genes as the most influential for prolonging overall survival (OS). To validate the significance of the *CHST4* gene in MPM, formalin-fixed paraffin-embedded (FFPE) specimens from our surgical cases were used to evaluate the intensity of expression by immunostaining, followed by statistical analysis for the association with survival. The results showed that the *CHST4* gene is involved in the immunological pathogenesis of MPM, a finding that may lead to the development of new prognostic prediction tools. CHST4, as a prognostic marker and being independent of the conventional TNM classification, can enable a more accurate assessment of tumor malignancy. Additionally, CHST4 is considered to be associated with tumor immunity against malignant tumors. Therefore, it may be helpful in determining the indications for surgery or immunotherapy in the case of MPM.

## 2. Results

### 2.1. Novel Classification of MPM Based on Comprehensive Survival Analysis of Cancer Genome Database

From the TCGA database, we extracted the clinical data and RNA-seq gene expression matrices of 85 patients with MPM. The patient backgrounds are presented in Appendix A. Among all samples, 81.2% were male and 97.6% were white. The proportion of epithelioid was 65.9%, and the numbers of patients with Stage I–II and III–IV were 25 (29.4%) and 60 (70.6%), respectively. The comprehensive multivariate Cox regression analysis of the gene expression of the TCGA’s MPM dataset for OS identified 1,246 prognostic genes (Appendix A). Consensus cluster analysis using the prognostic genes separated the patients into two groups (Figure 1A). Kaplan–Meier clustering analysis showed a difference in survival between these two groups (Figure 1B). We note that the observed difference may involve an overfitting bias caused by applying the supervised clustering (i.e., clustering following supervised feature selection) to the same data from which the clustering was developed. When these new clusters were compared with three histological subgroups of MPM, new clustering separated the patients in a manner different from that used to separate the conventional histological subgroups (Figure 1C). Although the prognosis was better for the epithelioid subgroup than for the histological subgroups, the patients in this subgroup were distributed evenly in a new clustering. Therefore, this novel classification based on the prognostic genes might reveal new biological features of MPM. Additionally, we investigated the cancer gene mutation in new clusters. Using the TCGA MPM dataset, the total number of patients with cancer gene mutation was determined (Appendix A), and five genes, including *NF2*, *BAP1*, *TP53*, *LATS2*, and *SETD2,* were found to be frequent. The mutations of the above five genes were counted in each new group (Appendix A). In cluster 2, there were more patients with no mutation, and patients with four mutations were included in only this cluster. We could not determine the relationship between mutation count and new clustering, and it was considered to be a novel classification to distinguish the patient prognosis regardless of the gene mutation.

### 2.2. Enrichment Analysis for Prognostic Genes of MPM

For the overall enrichment analysis of the top 300 favorable and unfavorable prognostic genes, we used Metascape, an effective program that leverages >40 different knowledgebases by combining functional enrichment, interactome analysis, gene annotation, and membership search [12]. Analysis of the worse prognostic genes showed that the highest-ranked statistical enrichment was associated with the GO mitotic cell cycle (Log_10_(*P*) = −67.03) (Figure 2A). On the other hand, for the better prognostic genes, the terms list showed that the highest-ranked enrichment was associated with the GO defense response to the virus (Log_10_(*P*) = −9.66) (Figure 2B). A pathway and process enrichment analysis was performed for every gene list, and terms with membership similarities and *p*-values < 0.01, minimum count of three, and enrichment factors > 1.5 (ratio between the observed counts and counts expected by chance) were clustered. For hierarchical clustering of enriched phases, kappa scores were used as the similarity metric, and subtrees with a similarity of >0.3 were regarded as clusters. The corresponding colors in Figure 2C,D reflect the independent clusters of various ontology keywords. We found eight clusters that refer to cell-cycle–related ontological terms in the worse prognostic gene analysis (Figure 2C) and five clusters that refer to immune-related ontological terms in the better prognostic gene analysis (Figure 2D). We then selected 10 highly expressed genes belonging to the mitotic cell cycle determined via unfavorable gene enrichment analysis and compared the expression of these genes between clusters (Appendix A). As a result, all 10 genes were significantly highly expressed in cluster 2. Moreover, we selected 10 highly expressed genes belonging to the defense response to the virus determined using favorable gene enrichment analysis and compared the expression of these genes between clusters (Appendix A). As a result, all genes except *MYD88* were significantly highly expressed in cluster 1. From these results, we assumed that MPMs with better prognoses possess immune-activated characteristics even if they were classified into different histological subtypes.

### 2.3. CHST4 as a Prognostic Gene of MPM and the Relationship with Immune-Related Gene Signatures

In the better prognostic genes, *CHST4* held the lowest hazard ratio (HR) in the comprehensive survival analysis (univariate: HR 0.37, *p* < 0.001; multivariate: HR 0.32, *p* < 0.001, Table 1). We investigated the relationship between *CHST4* mRNA expression in the MPM and Bindea immune-related gene signatures [13] representing 24 immune cell types. Three signatures, including CD8 naïve central memory cells (Tcm) (*p* = 0.037), Th17 cells (*p* = 0.004), and dendritic cells (DCs) (*p* = 0.014), were positively correlated with *CHST4* expression, and eight signatures, including T-helper cells (*p* = 0.006), Th2 cells (*p* < 0.001), T-follicular helper cells (TFHs) (*p* = 0.008), regulatory T cells (Tregs) (*p* = 0.033), natural killer cells (*p* = 0.005), NK56 bright cells (*p* < 0.001), plasmacytoid DC (pDC) (*p* = 0.001), and neutrophils (*p* = 0.003), were negatively correlated with *CHST4* expression (Figure 3A–K). We showed these immune-gene signatures and the relationship between *CHST4* expression and immune signatures in Figure 3L. From these results, we hypothesized that *CHST4* is useful as an immune and prognostic biomarker for MPM.

### 2.4. Evaluation Using MPM Tissue Samples of CHST4 Expression

Fifty-one patients who underwent extrapleural pneumonectomy (EPP) or pleurectomy/decortication (P/D) for MPM at Nagoya University Hospital between April 2006 and May 2021 were included (Figure 4). 

Of these 51 patients, those who failed to achieve macroscopic complete resection or had a history of respiratory disease or malignancy of other organs were excluded. Patients with Common Terminology Criteria for Adverse Events (CTCAE) Grade 3 or higher complications within 30 days after surgery were also excluded. In addition, immunohistochemistry (IHC) was performed on surgical specimens, but in one case, the CHST4 expression could not be scored because of low tumor-cell counts. Finally, 23 cases were included in this study, and the usability as a prognostic biomarker was validated by IHC.

Of the 23 patients included in the study, 19 (82.6%) were male, and the mean age was 60.7 years (Table 2). The histologic type was epithelioid mesothelioma in 18 patients (78.3%) and biphasic mesothelioma in 5 (21.7%), and the most common pathologic stage was stage I in 15 (65.2%) patients. EPP was performed in 15 (65.2%) patients and P/D in 8 (34.8%) patients. None of the patient backgrounds were significantly correlated with the intensity of *CHST4* expression (Appendix A).

To evaluate the intensity of CHST4 expression, we scored the IHC staining results as a percentage of CHST4 immunoreactivity in mesothelioma cells; since CHST4 localizes to the cytoplasm [8], the fraction of cytoplasmic CHST4-positive stained mesothelioma cells was scored after examining six high-power fields (40×) in one tissue section for each case (Figure 5A,B). The proportion and intensity of staining were scored (Table 3). The total score (TS) was the sum of the intensity + proportional scores. The cutoff for CHST4 strong expression was set according to the existing reports and the Kaplan–Meier curve and its Log-rank test for postoperative survival in the present study. The survival rates of the groups showed the greatest differences when the cutoff was CHST4-TS = 7 (Appendix A).

### 2.5. Statistical Analysis of CHST4 Protein Expression Intensity and Prognosis

The Kaplan–Meier method was used to calculate the postoperative survival rates. The OS curves for the two case groups with CHST4-TS = 7 as the cutoff are shown in Figure 6A; patients with CHST4-TS ≥ 7 had significantly better OS (*p* = 0.044), with a median postoperative survival of 107.8 months for CHST4-TS ≥ 7 and 38.0 months for CHST4-TS < 7. Postoperative recurrence-free survival was not significantly correlated with CHST4 expression (Appendix A), but survival after recurrence showed a trend similar to that of postoperative survival (Figure 6B), with a significantly better prognosis in cases with strong *CHST4* expression (*p* = 0.04).

Univariate and multivariate analysis of postoperative OS was performed (Table 4). In the univariate analysis, only pleural thickness (HR 1.02; 95% confidence interval (CI), 1.01 to 1.04; *p* = 0.01) was significantly associated with worse OS. Conversely, OS tended to improve in patients with strong *CHST4* expression (HR 0.23; 95% CI, 0.05 to 1.07; *p* = 0.06). In multivariate analysis, only pStage was a significant risk factor (HR 3.06; 95% CI, 1.48 to 6.33; *p* = 0.003). The expression of *CHST4* tended to be associated with improved OS, with the lowest hazard and marginal significance (HR 0.12; 95% CI, 0. 01 to 1.15; *p* = 0.06) adjusted by age, sex, pStage, histological type, and pleural thickness.

## 3. Discussion

Traditionally, chemotherapy and radiation therapy have been used for unresectable MPM and for resectable lesions as part of multimodality treatment. The combination of cisplatin and pemetrexed has been the preferred chemotherapy in most cases. However, the OS for chemotherapy alone is 12.1 months [14], and the OS for multimodality treatment that includes surgery for resectable MPM is 16.8 to 30 months, which is unsatisfactory [15,16,17,18]. Recently, the results of the MARS 2 trial, which compared the effectiveness and cost-effectiveness of surgery versus no surgery for the treatment of MPM, were presented, and there was no significant difference in survival time between the two groups. Moreover, the risk of grade 3 or higher adverse events was greater for patients in the surgery arm than for those in the chemotherapy-alone arm [19]. Consequently, there is an increasing need for biomarkers to predict postoperative prognosis and carefully select the therapeutic strategy for each patient with MPM.

There have been emerging reports of the efficacy of immunotherapy for MPM as a new treatment option superior to chemotherapy, and a phase III trial compared the results of nivolumab plus ipilimumab with those of conventional chemotherapy in patients with MPM and Performance Scores from 0 to 1. In this trial, the combination of nivolumab and ipilimumab resulted in better OS (18.1 months vs. 14.1 months, HR 0.74, *p* = 0.002) and a better 2-year OS rate (41% vs. 27%) [20]. Moreover, a phase II trial of neoadjuvant durvalumab versus durvalumab plus tremelimumab followed by surgery in patients with MPM was also reported, and the median OS was longer for the patients receiving combined immunotherapy than for the patients receiving monotherapy or no immunotherapy [21]. As immunotherapy has an increasingly important role as a treatment option for MPM, it is desirable to develop predictors of treatment response to better select patients.

In the past decade, genomic studies uncovered molecular profiles related to the histopathological classifications of MPM, which included epithelioid, biphasic, and sarcomatoid, and each was enriched with somatic alterations in known cancer genes [22,23]. For example, epithelial and sarcomatoid MPM harbored BAP1 and TP53 mutations, respectively. A recent multiomics study showed that the current histopathological classification of MPM only explains a fraction of the molecular heterogeneity of the disease and that ploidy, adaptive immune response, and CpG island methylation were similarly important [24]. In our study, we performed transcriptome analysis that primarily focused on the relationship between survival and gene expression and better prognostic MPMs expressed immune-activated characteristics regardless of the histology. To take advantage of this feature in clinical diagnosis, we selected *CHST4*, which was identified as the most influential gene for prolonged OS, and created a diagnostic IHC method.

*CHST4* is a sulfotransferase expressed in HEVs [8] and has an important role in the immune system, given that it is involved in homing of lymphocytes in lymphoid organs [25]. It is assumed that *CHST4* is involved in tumor immunity against malignant tumors, and functional analysis of *CHST4* is expected to contribute to the development of immunotherapy. Many reports have revealed the function of *CHST4* in malignant tumors, most frequently in hepatocellular carcinoma, suggesting that genetic background has a role in the response rate after immunotherapy. *CHST4* has been identified as a predictor of treatment response to immunotherapy and is known to have a positive effect on treatment response [26]. In the case of triple-negative breast cancer, *CHST4* expression has also been identified as a prognostic factor when effective treatment options are limited. The other four genes, *COCH*, *CST9*, *SOX11*, and *TDGF1*, as well as *CHST4*, have also been implicated in the immune-related GO term, indicating the importance of immunomodulatory mechanisms in cancer therapy [27].

Immunotherapy is becoming increasingly important in MPM, and it is crucial to clarify the relationship between immunomodulatory mechanisms, including *CHST4*, and prognosis and therapeutic efficacy. In this study, we used tissue samples from patients with MPM who underwent surgery at Nagoya University Hospital to investigate the relationship between *CHST4* expression and prognosis. The OS tended to be better in patients with *CHST4* expression, and the survival rate after postoperative recurrence was also better. Because only 6 patients out of 23 were treated with immunotherapy and the details of treatment before and after immunotherapy (chemotherapy and radiotherapy) differed among the patients, the response to immunotherapy could not be statistically evaluated. Although *CHST4* is not a gene that is specifically expressed in MPM and is limited to cases in which a definitive diagnosis has been obtained via biopsy or other means, *CHST4* may be involved in tumor immunity to MPM and can be used to predict therapeutic efficacy and prognosis.

Univariate and multivariate analysis of the 23 patients included in the study suggested that pathological stage and pleural thickness also influenced postoperative OS. Pleural thickness is defined as the thoracic cavity thickness measured vertically in three dimensions on computed tomography images. The International Association for the Study of Lung Cancer has noted that patients with thicker pleural thickness have poorer OS [28], and the pleural thickness is probably included in the T classification of MPM in the 9th edition of the TNM classification as a prognostic factor [28,29]. In addition, pleural thickness is a predictor of postoperative outcome. Pleural thickness was also reportedly a risk factor for postoperative complications [30] and may reflect the overall disease status of MPM, but it did not correlate with the intensity of *CHST4* expression. If the *CHST4* expression intensity is considered to be a prognostic factor, which is independent of the conventional TNM classification, their combination should provide a more accurate assessment of the disease.

One limitation of this study was the limited number of cases. The number of cases of MPM for which surgery is indicated is small, and they require highly invasive surgery, which is associated with a high risk of severe postoperative complications. In addition, to ensure a more accurate prognostic analysis, patients with CTCAE Grade 3 or higher complications were excluded, resulting in a smaller number of cases.

However, this study is unique in that it targets *CHST4*, whose role in MPM has not previously been studied. Although there are several studies that investigated useful markers for the diagnosis and prognosis of MPM [31,32], the uniqueness of this study lies in the identification of the *CHST4* gene, which is a prognostic factor in MPM, and further investigation of the immunostaining scoring. This demonstrated the possibility of application of *CHST4* to real-world clinical practice. It is expected that studies using even more cases will be conducted in the future.

## 4. Materials and Methods

### 4.1. Discovery of Prognostic Genes and Consensus Clustering

Multivariate Cox regression analysis was used to evaluate the association between OS and the expression of each gene in the TCGA MPM dataset. Specifically, Cox regression models with each gene expression level, gender, age, and pathologic tumor stage as explanatory variables and OS as response variables were fitted separately for each gene. From each estimated model, a p-value was calculated to test the gene association, which was then used in subsequent screening. For screening prognostic genes, the false discovery rate that represents the percentage of incorrectly extracted genes among all extracted genes was controlled to <1%. The R script for multivariate Cox regression analysis in Section 4.1 is provided in Appendix A. Consensus clustering [33] was then applied to these prognostic genes to identify the clusters without using outcome information. Cluster labels were obtained by aggregating the results of hierarchical clustering based on Pearson correlation analysis of 1000 subsamples, which used 80% of the samples each time for different numbers of clusters (K) by varying K from 2 to 10. The number of clusters was determined to achieve the most consistent clustering results [33].

### 4.2. Overall Enrichment Analysis

The Metascape program was used to conduct a functional analysis of lists of prognostic gene ENTREZ IDs. Metascape leverages > 40 separate knowledge bases by combining functional enrichment, interactome analysis, gene annotation, and membership search [34]. The ontology sources of the KEGG Pathway, GO Biological Processes, Reactome Gene Sets, Canonical Pathways, CORUM, WikiPathways, and PANTHER Pathway were utilized to identify all statistically enriched terms. Accumulative hypergeometric *p*-values and enrichment factors were then computed and used for filtering. The remaining significant terms were then grouped into a tree in a hierarchical fashion using Kappa statistical similarities between their gene memberships (similar to the method used in the NCI DAVID site). Subsequently, a threshold of 0.3 for the kappa score was used to group the terms in the tree into clusters. A subset of representative terms that were chosen from the entire cluster was used to create a network layout. More precisely, each term was represented by a circle node, the color of which indicates the cluster identification and the size of which is correlated with the number of input genes falling under that term. An edge connects terms with a similarity score > 0.3; the edge’s thickness indicates the similarity score. Using Cytoscape, the network is shown with a “force-directed” structure with edge bundling for clarity. A single term is chosen from every cluster, and its label is the term description.

### 4.3. Calculation of the Immune-Gene Signature

Cell-type enrichment analysis was performed by use of xCell, “https://xcell.ucsf.edu/ (accessed on 5 October 2023)” [35], and the Bindea immune-gene signature [13] was applied to analyze the status of anticancer immunity and the proportion of tumor-infiltrating immune cells [36]. Pearson’s correlation was applied to calculate the correlation between each gene expression and the immune activity score from cell-type enrichment analysis.

### 4.4. Patient Data and Tumor Material

The patients underwent surgery for MPM at the Department of Thoracic Surgery, Nagoya University Hospital, from April 2006 to May 2021 and were diagnosed by the Department of Pathology at the same hospital. All clinical data were obtained from medical records. To purely compare the nature and prognosis of MPM as a malignant disease, we excluded patients with a history of respiratory disease or malignancy of other organs and patients with CTCAE grade 3 or higher complications within 30 days after surgery. FFPE samples were prepared and used for the *CHST4* expression analysis.

### 4.5. Immunohistochemistry and Evaluation of Stained Slides

Four-micron-thick sections of FFPE tissue were deparaffinized, subjected to antigen retrieval using a pressure cooker, and immunohistochemically stained with anti-CHST4 antibody (rabbit polyclonal, 1:50, HPA021955, Sigma-Aldrich, St. Louis, Missouri, USA). CHST4 protein expression was assessed by the investigators, including one pathologist, blinded to the patient’s information. Staining intensity was evaluated by assigning a score consisting of the sum of the percentage of positive cells (0–5) and the staining intensity of the positive cells (0–3). The proportion of positive cells (Proportion Score, PS) was evaluated as follows: PS = 0 if no positive cells were found, PS = 1 if positive cells were less than 1/100, PS = 2 if 1/100 to 1/10, PS = 3 if 1/10 to 1/3, PS = 4 if 1/3 to 2/3. The Intensity Score (IS) was evaluated as follows: IS = 0 for negative, IS = 1 for weakly positive, IS = 2 for moderately positive, and IS = 3 for strongly positive. The TS is the sum of PS and IS. The cutoff was set at TS = 7 because the survival rates differed significantly. For reference, a scale that uses a similar method to evaluate staining intensity is the Allred Score. It is also used to evaluate the expression of hormone receptors (estrogen receptor, progesterone receptor) in breast cancer, and although there are no clear criteria, some studies set the cutoff for strong positivity at TS = 7 [37,38,39,40]. The Allred score for breast cancer is an evaluation method that has already been used in clinical practice, and we consider that it can be used to evaluate gene expression in MPM with the same level of reliability.

### 4.6. Statistical Analysis

R software version 4.1.1 was used for all statistical analysis of the TCGA MPM dataset. We used the “survival” package [41,42] for the Cox regression, the “qvalue” package [43] for the false discovery control, and the “ConsensusClusterPlus” package [44] for the Consensus Clustering in R. For statistical analysis of cases operated upon at our hospital, we used EZR software version 1.54 [45]. The *t*-test was performed to assess the presence of patient background bias, such as age, sex, and histological type. To compare survival between groups on the basis of staining scores from immunohistochemical staining, Kaplan–Meier OS analysis curves were generated and compared by performing the log-rank test. Univariate and multivariate Cox proportional hazards regression analyses were also performed to estimate HRs and 95% CIs for the HRs. Statistical significance was set at *p* < 0.05 (two-sided).

## 5. Conclusions

The *CHST4* expression intensity was found to be a potential postoperative MPM prognostic factor, and *CHST4* can possibly be used to predict postoperative prognosis.

## Figures and Tables

**Figure 1 ijms-25-02270-f001:**
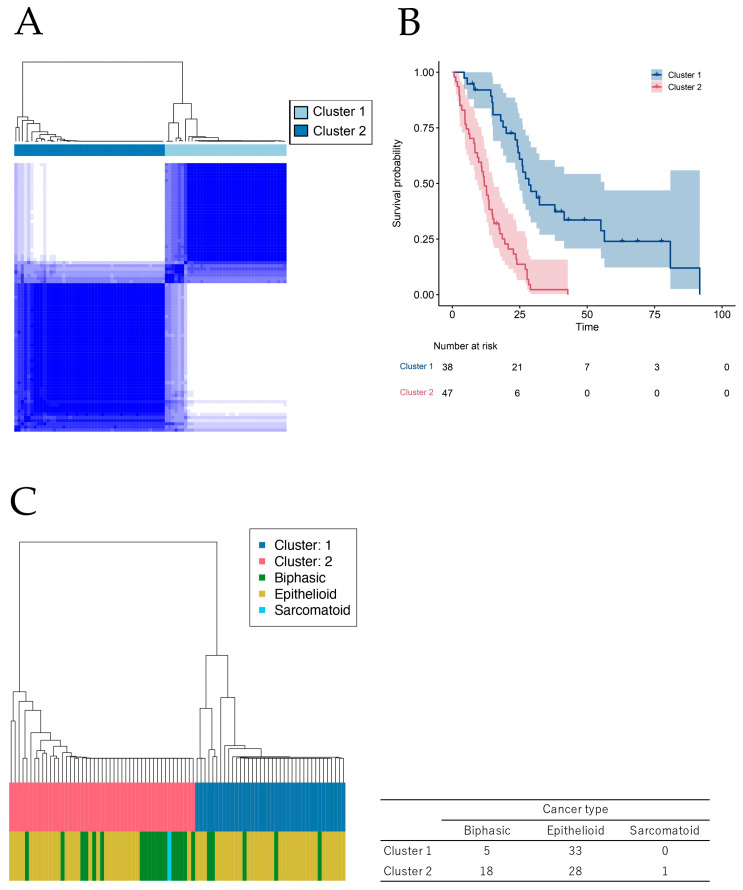
Survival analysis across the entire transcriptome and new clusters unrelated to previous histological subtypes. (**A**) Consensus clustering using prognostic genes derived from screening by multiple testing with multivariate Cox regression analysis of gene expression in The Cancer Genome Atlas (TCGA) malignant pleural mesothelioma (MPM) dataset for OS. (**B**) Kaplan–Meier analysis using the TCGA MPM dataset was performed for two patient groups derived from consensus clustering. (**C**) The comparison between the conventional histological subtypes and new patient groups derived from the consensus clustering. The right-side table shows the patient numbers for each cluster and the conventional histological type.

**Figure 2 ijms-25-02270-f002:**
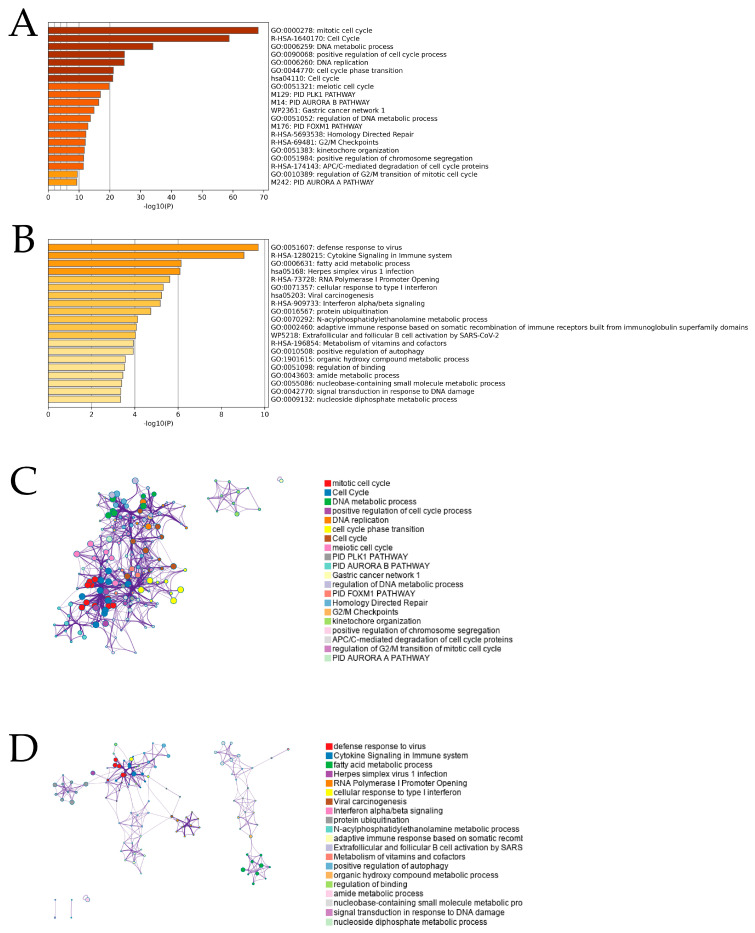
Metascape functional analysis for unfavorable and favorable prognostic genes. The bar graph shows the top 20 clusters of enriched terms associated with unfavorable genes (**A**) and favorable genes (**B**). The color indicates the size of the *p*-value and the probability that the null hypothesis is rejected. The graph indicates the network of enriched terms associated with unfavorable genes (**C**) and favorable genes (**D**). The nodes are colored by cluster ID, and the nodes that share the same cluster ID are typically close to each other.

**Figure 3 ijms-25-02270-f003:**
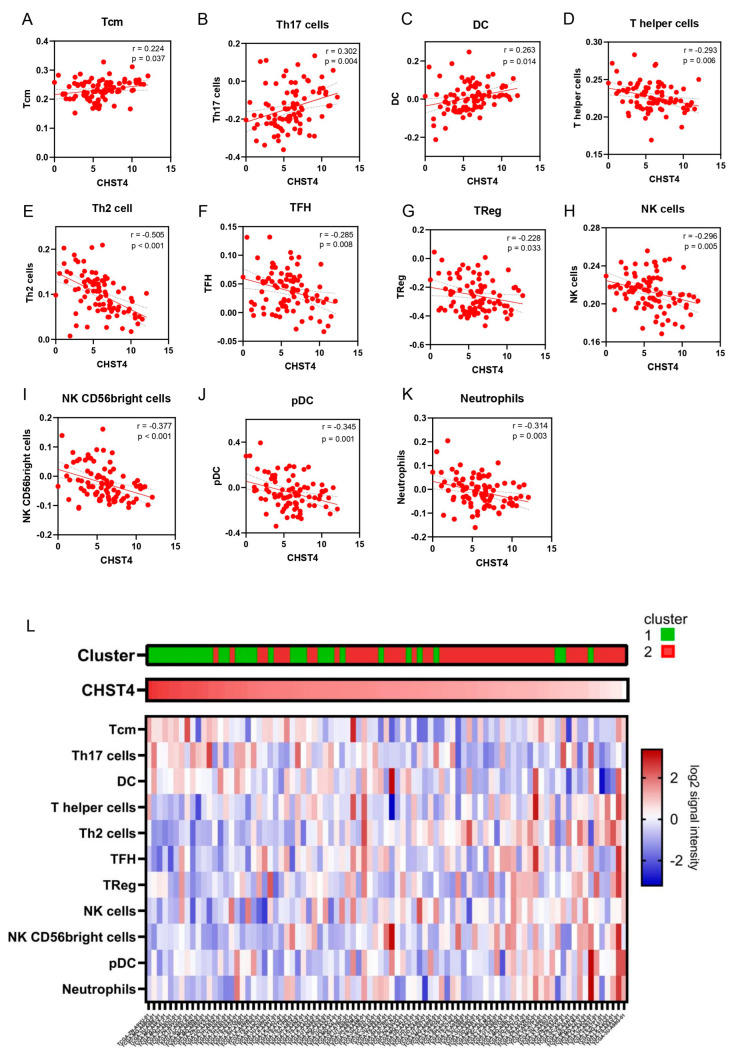
Relationship between *CHST4* expression and Bindea immune−cell gene signatures. (**A**–**K**) The scatter plot shows the correlation between *CHST4* expression and each immune−cell gene signature. Immune−cell gene signatures with significant correlations are indicated. (**L**) Heatmap depicting the relationship among clusters derived from comprehensive survival analysis, *CHST4* expression, and the immune−cell gene signatures shown in (**A**–**K**) above. The samples are sorted by *CHST4* expression.

**Figure 4 ijms-25-02270-f004:**
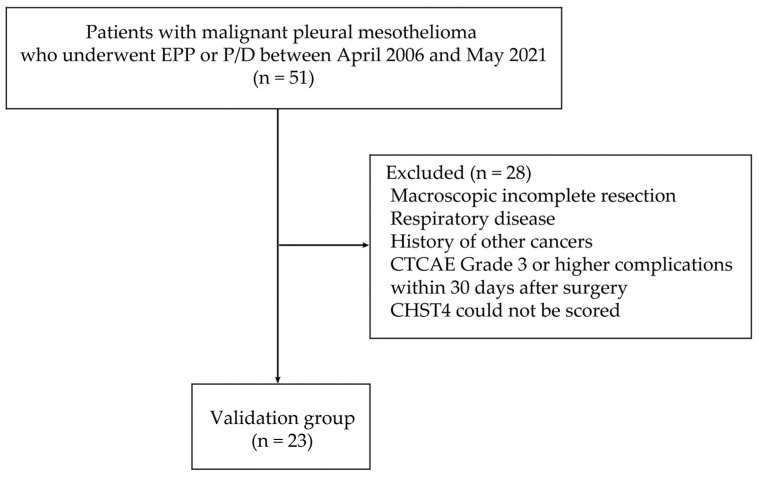
Flow chart of patient selection. Of 51 patients who underwent surgery for malignant pleural mesothelioma at Nagoya University Hospital, 23 were included in this study, and the usability as a prognostic biomarker was validated by immunohistochemistry.

**Figure 5 ijms-25-02270-f005:**
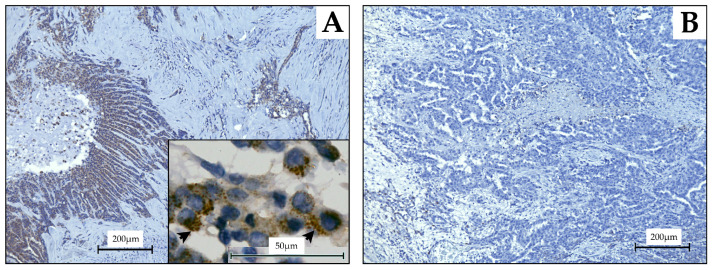
Images of immunohistochemistry performed on malignant pleural mesothelioma specimens. In case (**A**), which had a relatively good prognosis with a postoperative survival of 61.4 months, CHST4 immunoreactivity was localized in the perinuclear cytoplasm (indicated by arrowhead). (**B**) On the other hand, in the case with a poor prognosis with a postoperative survival of 7.6 months, no CHST4 expression was observed.

**Figure 6 ijms-25-02270-f006:**
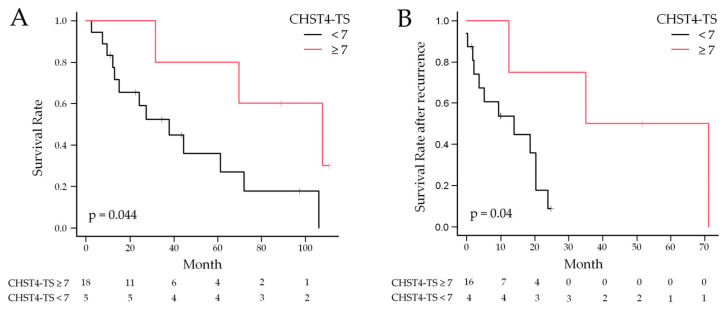
Kaplan–Meier survival curves for evaluating postoperative overall survival (OS) (**A**) and OS after recurrence (**B**) in two groups divided by the CHST4-TS. Patients with CHST4-TS ≥ 7 had significantly better survival (107.8 months vs. 38.0 months), and survival after recurrence also showed a similar trend (53.2 months vs. 14.0 months).

**Table 1 ijms-25-02270-t001:** Top 20 favorable prognostic genes detected by univariate and multivariate Cox regression.

Univariate Analysis			Multivariate Analysis		
Hugo Symbol	Hazard Ratio (95% CI)	*p*-Value	Hugo Symbol	Hazard Ratio (95% CI)	*p*-Value
CHST4	0.37 (0.21–0.65)	0.0007	FOXO4	0.22 (0.09–0.52)	0.0006
EMB	0.38 (0.23–0.61)	<0.0001	CACHD1	0.31 (0.18–0.53)	<0.0001
CACHD1	0.39 (0.24–0.63)	0.0002	CHST4	0.32 (0.17–0.60)	0.0003
EPB41L4A	0.41 (0.25–0.66)	0.0002	EMB	0.35 (0.21–0.58)	<0.0001
ATP8A1	0.42 (0.26–0.68)	0.0004	PRIMA1	0.36 (0.20–0.66)	0.0008
GHR	0.42 (0.27–0.64)	<0.0001	GHR	0.36 (0.23–0.56)	<0.0001
EMBP1	0.42 (0.28–0.64)	<0.0001	PLEKHH1	0.39 (0.24–0.63)	0.0001
TNFSF13	0.43 (0.30–0.60)	<0.0001	EMBP1	0.39 (0.25–0.61)	<0.0001
BTN3A3	0.44 (0.30–0.65)	<0.0001	EPB41L4A	0.40 (0.25–0.64)	0.0002
HIST1H2AC	0.45 (0.31–0.64)	<0.0001	ADH1B	0.41 (0.25–0.68)	0.0005
RTP4	0.45 (0.31–0.66)	<0.0001	THTPA	0.41 (0.29–0.59)	<0.0001
KLHL9	0.45 (0.34–0.59)	<0.0001	HIST1H2AC	0.41 (0.28–0.58)	<0.0001
THTPA	0.46 (0.32–0.65)	<0.0001	RICH2	0.41 (0.28–0.61)	<0.0001
NUDT7	0.46 (0.32–0.67)	<0.0001	TNFSF13	0.41 (0.29–0.58)	<0.0001
HIST1H2BD	0.46 (0.32–0.67)	<0.0001	KLHL9	0.41 (0.30–0.55)	<0.0001
C5orf4	0.47 (0.33–0.66)	<0.0001	BTN3A3	0.42 (0.28–0.63)	<0.0001
FLJ11235	0.47 (0.34–0.65)	<0.0001	ATP8A1	0.42 (0.25–0.69)	0.0006
DCAF11	0.47 (0.35–0.63)	<0.0001	RTP4	0.42 (0.29–0.61)	<0.0001
TMCO4	0.47 (0.35–0.63)	<0.0001	HIST1H2BC	0.42 (0.26–0.67)	0.0003
SH3BGRL	0.47 (0.35–0.64)	<0.0001	HIST1H2BD	0.42 (0.29–0.60)	<0.0001

**Table 2 ijms-25-02270-t002:** Clinicopathological characteristics of malignant pleural mesothelioma patients with CHST4 immunohistochemistry.

Clinicopathological Feature		n = 23
Sex		
	Male	19 (82.6%)
	Female	4 (17.4%)
Age, years (median, range)		61 (46–73)
Histological type		
	Epithelioid mesothelioma	18 (78.3%)
	Biphasic mesothelioma	5 (21.7%)
Pleural thickness, mm (median, range)		29.4 (11.1–128.8)
Surgical procedure		
	Extrapleural pneumonectomy	15 (65.2%)
	Pleurectomy/decortication	8 (34.8%)
pStage		
	I	15 (65.2%)
	II	0 (0.0%)
	III	7 (30.4%)
	IV	1 (4.3%)

**Table 3 ijms-25-02270-t003:** Scoring criteria for CHST4 staining.

Proportion of Positively Stained at Perinuclear Cytoplasm	Proportion Score (PS)	Average Intensity of Positively Stained at Perinuclear Cytoplasm	Intensity Score (IS)
None	0	None	0
<1/100	1	Weak	1
1/100 to 1/10	2	Moderate/medium	2
1/10 to 1/3	3	Strong	3
1/3 to 2/3	4		
>2/3	5		

**Table 4 ijms-25-02270-t004:** Univariate and multivariate Cox regression analyses of overall survival in malignant pleural mesothelioma patients.

Variable	Univariate Analysis				Multivariate Analysis			
	HR	95% CI		*p*-Value	HR	95% CI		*p*-Value
		Lower	Upper			Lower	Upper	
Age	0.98	0.93	1.05	0.65	1.00	0.91	1.10	0.98
Sex (male)	2.26	0.62	8.20	0.22	3.91	0.51	29.8	0.19
pStage	1.47	0.97	2.22	0.07	3.06	1.48	6.33	0.003
Histological type (epithelioid)	0.46	0.16	1.36	0.16	0.67	0.15	3.01	0.60
Pleural thickness	1.02	1.01	1.04	0.01	1.01	0.99	1.03	0.28
CHST4-TS (≥7)	0.23	0.05	1.07	0.06	0.12	0.01	1.15	0.06

Abbreviations: HR—hazard ratio, CI—confidence interval. TNM stage is based on AJCC 8th edition.

## Data Availability

The datasets used during the present study are available from the corresponding author upon reasonable request.

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
