# Peer review of "CHST4 Gene as a Potential Predictor of Clinical Outcome in Malignant Pleural Mesothelioma"

_ijms, 2024, doi:10.3390/ijms25042270_

Round 1

Reviewer 1 Report

Comments and Suggestions for Authors

This is a great manuscript.

The authors can still improve some parts of it:

1. This part of the methods requires more explanation: "Multivariate Cox regression analysis adjusted for sex, age, and pathological tumor  stage was used to evaluate the association between OS and the expression of each gene in the TCGA MPM dataset. For the screening of prognostic genes, the false discovery rate, which represents the percentage of incorrectly extracted genes among all extracted 316 genes, was controlled to < 1%."

Could authors share the R script or explain more, how did they applied adjusted regression model on each gene? Was it gene-wise or was it some general expression measure? This is fundamental part fo the apaer and needs more explanation for the readers.

2. This study considered only RNAseq data, but I encourage authors also to include WGS/WES or any DNA data available. Thsi way they can build a more comprehensive analysis, like in thsia article PMID: 25496518. At least, this option should be discussed by the authors -combination of DNA and RNA seq data would increase the poser of analysis and helps to explain the molecular machinery of the tumors.

3. What were these 1246 prognostic geens and what were their expression values/differences? I couldn't find any table (supplementary) with these data. The authors should add supplementary table describing all 1246 genes.

4. The pathway analysis is a little superficial. Here is one published work, PMID: 18514490, that the authors can study and follow in order to explain the genetic pathways in their functional context. Metascape analysis is nice, but is is based on canonical pathways- it means that if you have 50% of your genes in one pathway and 50% in another, then you get hit in two pathways. But maybe these all genes, 100% belong to some pathway that is not canonical. I suggest the authors analysegenes in their functional context, lie was in the PMID: 18514490.

Author Response

1. This part of the methods requires more explanation: "Multivariate Cox regression analysis adjusted for sex, age, and pathological tumor  stage was used to evaluate the association between OS and the expression of each gene in the TCGA MPM dataset. For the screening of prognostic genes, the false discovery rate, which represents the percentage of incorrectly extracted genes among all extracted 316 genes, was controlled to < 1%."
Could authors share the R script or explain more, how did they applied adjusted regression model on each gene? Was it gene-wise or was it some general expression measure? This is fundamental part of the apear and needs more explanation for the readers.

Reply: Thank you so much for your question. This analysis was basically gene-wise analysis. We provided the R script for this analysis and modified the method description (Line 338-343).

2. This study considered only RNAseq data, but I encourage authors also to include WGS/WES or any DNA data available. Thsi way they can build a more comprehensive analysis, like in thsia article PMID: 25496518. At least, this option should be discussed by the authors -combination of DNA and RNA seq data would increase the poser of analysis and helps to explain the molecular machinery of the tumors.

Reply: Thank you for your important suggestion. We added the analysis of cancer gene mutation and realized that cancer gene mutation was not correlated with our new clustering based on RNA-seq analysis. Therefore, we understood this new clustering and CHST4 was novel biomarker to distinguish the poor prognosis patients regardless of gene mutation. We modified Line 100-107 in the manuscript and added Figure S1 and S2.

3. What were these 1246 prognostic genes and what were their expression values/differences? I couldn't find any table (supplementary) with these data. The authors should add supplementary table describing all 1246 genes.

Reply: As following your suggestion, we added the 1246 prognostic gene list as Table S2.

4. The pathway analysis is a little superficial. Here is one published work, PMID: 18514490, that the authors can study and follow in order to explain the genetic pathways in their functional context. Metascape analysis is nice, but is is based on canonical pathways- it means that if you have 50% of your genes in one pathway and 50% in another, then you get hit in two pathways. But maybe these all genes, 100% belong to some pathway that is not canonical. I suggest the authors analysegenes in their functional context, lie was in the PMID: 18514490.

Reply: We appreciate for your valuable opinion. We additionally analyzed top unfavorable GO mitotic cell cycle and favorable GO defense response to virus. As a result, we realized top belonging genes in mitotic cell cycle were significantly higher expressed in cluster 2 and top belonging genes in defense response to virus were significantly higher expressed in cluster 1. Therefore, these GO were probably related to the prognosis of MPM patients. We modified Line 135-142 and added Figure S3 and S4.

Reviewer 2 Report

Comments and Suggestions for Authors

This paper utilized the Cancer Genome Atlas for survival analysis and identified the CHST4 gene as a potential predictor of overall survival in patients with MPM and validated the relationship between CHST4 expression and prognosis. It confirms that CHST4 can be used as a prognostic biomarker for MPM. Generally, this is a well-written paper, but some issues should be properly addressed. I will recommend this work to be published after minor revision in this journal.

1.        It is recommended that the clinical significance of CHST4 as a potential prognostic biomarker be discussed in the introduction. How does this information apply to clinical practice? What impact might it have on patient outcomes or treatment strategies?

2.        In the introduction, lines 67-69, does the fact that CHST4 can also be used as a marker for other tumors mean that CHST4 is a generalized tumor marker and not MPM-specific? If so, the authors should point this out in the conclusion and suggest in the outlook what modalities can improve the specificity of CHST4 for diagnosing MPM by combining it.

3.        Focusing on clarity and simplicity of expression would enhance overall readability. It is recommended that authors avoid the use of long, complex sentences. For example, in lines 25-28.

4.        It is recommended that the article be supplemented with specific criteria for patient selection.

5.        It is recommended that a description of comparisons with other relevant studies be added to the text to assess the sophistication and uniqueness of its findings.

6.        For Figure 3L, unreadable stacked sample numbers on the bottom.

Author Response

1. It is recommended that the clinical significance of CHST4 as a potential prognostic biomarker be discussed in the introduction. How does this information apply to clinical practice? What impact might it have on patient outcomes or treatment strategies?

Reply: Thank you for your interest in our report. The following description has been added to the Introduction (Line 75-79).
CHST4, as a prognostic marker and being independent of the conventional TNM classification, can enable more accurate assessment of tumor malignancy. Additionally, CHST4 is considered to be associated with tumor immunity against malignant tumors. Therefore, it may be helpful in determining the indications for surgery or immunotherapy in case of MPM.

2. In the introduction, lines 67-69, does the fact that CHST4 can also be used as a marker for other tumors mean that CHST4 is a generalized tumor marker and not MPM-specific? If so, the authors should point this out in the conclusion and suggest in the outlook what modalities can improve the specificity of CHST4 for diagnosing MPM by combining it.

Reply: Thank you for pointing out the tumor specificity of CHST4. The findings presented in this report are mainly that the expression of CHST4 may be used as a prognostic marker after surgical treatment of MPM and may predict the efficacy of immunotherapy. In principle, these findings are to be evaluated in cases in which a definitive diagnosis of MPM has been obtained by pleural biopsy or other means, so we do not envision situations that require comparison with other malignancies. Of course, we do not anticipate the use of CHST4 as a general tumor marker for screening. The point that CHST4 is not a MPM-specific gene should be made clear in the paper and is mentioned in Discussion (Line 307-310).

3. Focusing on clarity and simplicity of expression would enhance overall readability. It is recommended that authors avoid the use of long, complex sentences. For example, in lines 25-28.

Reply: As you pointed out, the text was complicated in some places. We have changed the complicated structure of the text as much as possible, including the part you pointed out.

4. It is recommended that the article be supplemented with specific criteria for patient selection.

Reply: We added details to 4.4. Patient data and tumor material in Materials and Methods (Line 378-384).

5. It is recommended that a description of comparisons with other relevant studies be added to the text to assess the sophistication and uniqueness of its findings.

Reply: For the evaluation of IHC staining intensity in MPM, the Allred score is used as a reference.
As described in 4.5. Immunohistochemistry and evaluation of stained slides in Materials and Methods, the Allred score is mainly used to evaluate hormone receptor expression in breast cancer. It is reliable enough to be used at the actual clinical level. This information has been added to Materials and Methods (Line 401-403).
This report is unique in that it targets CHST4, whose role in MPM has not been investigated before, and in that CHST4 may be involved in tumor immunity and may be a useful tool to carry out immunotherapy, which will become more important in the future. The above information has been added to the Discussion section along with the cited references (Line 329-335).

6. For Figure 3L, unreadable stacked sample numbers on the bottom.

Reply: Thanks for pointing this out, we have corrected Figure 3L.

Round 2

Reviewer 1 Report

Comments and Suggestions for Authors

All my comments have been answered.